# The challenges arising from the COVID-19 pandemic and the way people deal with them. A qualitative longitudinal study

Dominika Maison[☉], Diana Jaworska[☉], Dominika Adamczyk[iD][☉]*, Daria Affeltowicz[iD][☉]

Faculty of Psychology, University of Warsaw, Warsaw, Poland

☉ These authors contributed equally to this work.
* dominika.adamczyk@psych.uw.edu.pl

**Data Availability Statement:** All relevant data are within the manuscript and its Supporting Information files (S1 Dataset).

## Abstract

The conducted qualitative research was aimed at capturing the biggest challenges related to the beginning of the COVID-19 pandemic. The interviews were carried out in March-June (five stages of the research) and in October (the 6th stage of the research). A total of 115 in-depth individual interviews were conducted online with 20 respondents, in 6 stages. The results of the analysis showed that for all respondents the greatest challenges and the source of the greatest suffering were: a) limitation of direct contact with people; b) restrictions on movement and travel; c) necessary changes in active lifestyle; d) boredom and monotony; and e) uncertainty about the future.

## Introduction

The coronavirus disease (COVID-19), discovered in December 2019 in China, has reached the level of a pandemic and, till June 2021, it has affected more than 171 million people worldwide and caused more than 3.5 million deaths all over the world [1]. The COVID-19 pandemic as a major health crisis has caught the attention of many researchers, which has led to the creation of a broad quantitative picture of human behavior during the coronavirus outbreak [2–4]. What has been established so far is, among others, the psychological symptoms that can occur as a result of lockdown [2], and the most common coping strategies [5]. However, what we still miss is an in-depth understanding of the changes in the ways of coping with challenges over different stages of the pandemic. In the following study, we used a longitudinal qualitative method to investigate the challenges during the different waves of the coronavirus pandemic as well as the coping mechanisms accompanying them.

In Poland, the first patient was diagnosed with COVID-19 on the 4th March 2020. Since then, the number of confirmed cases has grown to more than 2.8 million and the number of deaths to more than 73,000 (June 2021) [1]. From mid-March 2020, the Polish government, similarly to many other countries, began to introduce a number of restrictions to limit the spread of the virus. These restrictions had been changing from week to week, causing diverse reactions in people [6]. It needs to be noted that the reactions to such a dynamic situation cannot be covered by a single study. Therefore, in our study we used qualitative longitudinal

**Funding:** This work was supported by the Faculty of Psychology, University of Warsaw, Poland from the funds awarded by the Ministry of Science and Higher Education in the form of a subsidy for the maintenance and development of research potential in 2020 (501-D125-01-1250000). The funders had no role in study design, data collection and analysis, decision to publish, or preparation of the manuscript.

**Competing interests:** The authors have declared that no competing interests exist.

research in order to monitor changes in people's emotions, attitudes, and behavior. So far, few longitudinal studies have been carried out that investigated the various issues related to the COVID-19 pandemic; however, all of them were quantitative [7–10]. The qualitative approach (and especially the use of enabling and projective techniques) allows for an in-depth exploration of respondents' reactions that goes beyond respondents' declarations and captures what they are less aware of or even unconscious of. This study consisted of six stages of interviews that were conducted at key moments for the development of the pandemic situation in Poland. The first stage of the study was carried out at the moment of the most severe lockdown and the biggest restrictions (March 2020) and was focused on exploration how did people react to the new uncertain situation. The second stage of the study was conducted at the time when restrictions were extended and the obligation to cover the mouth and nose everywhere outside the household were introduced (middle of April 2020) and was focused at the way how did people deal with the lack of family gatherings over Easter. The third stage of the study was conducted at the moment of announcing the four stages of lifting the restrictions (April 2020) and was focused on people's reaction to an emerging vision of getting back to normalcy. The fourth stage of the study was carried out, after the introduction of the second stage of lifting the restrictions: shopping malls, hotels, and cultural institutions were gradually being opened (May 2020). The fifth stage of the study was conducted after all four stages of restriction lifting were in place (June 2020). Only the obligation to cover the mouth and nose in public spaces, an order to maintain social distance, as well as the functioning of public places under a sanitary regime were still in effect. During those 5 stages coping strategies with the changes in restrictions were explored. The sixth and last stage of the study was a return to the respondents after a longer break, at the turn of October and November 2020, when the number of coronavirus cases in Poland began to increase rapidly and the media declared "the second wave of the pandemic". It was the moment when the restrictions were gradually being reintroduced. A full description of the changes occurring in Poland at the time of the study can be found in S1 Table.

The following study is the first qualitative longitudinal study investigating how people cope with the challenges arising from the COVID-19 pandemic at its different stages. The study, although conducted in Poland, shows the universal psychological relations between the challenges posed by the pandemic (and, even more, the restrictions resulting from the pandemic, which were very similar across different countries, not only European) and the ways of dealing with them.

## Literature review

The COVID-19 pandemic has led to a global health crisis with severe economic [11], social [3], and psychological consequences [4]. Despite the fact that there were multiple crises in recent years, such as natural disasters, economic crises, and even epidemics, the coronavirus pandemic is the first in 100 years to severely affect the entire world. The economic effects of the COVID-19 pandemic concern an impending global recession caused by the lockdown of non-essential industries and the disruption of production and supply chains [11]. Social consequences may be visible in many areas, such as the rise in family violence [3], the ineffectiveness of remote education, and increased food insecurity among impoverished families due to school closures [12]. According to some experts, the psychological consequences of COVID-19 are the ones that may persist for the longest and lead to a global mental health crisis [13]. The coronavirus outbreak is generating increased depressive symptoms, stress, anxiety, insomnia, denial, fear, and anger all over the world [2, 14]. The economic, social, and psychological

problems that people are currently facing are the consequences of novel challenges that have been posed by the pandemic.

The coronavirus outbreak is a novel, uncharted situation that has shaken the world and completely changed the everyday lives of many individuals. Due to the social distancing policy, many people have switched to remote work—in Poland, almost 75% of white-collar workers were fully or partially working from home from mid-March until the end of May 2020 [15]. School closures and remote learning imposed a new obligation on parents of supervising education, especially with younger children [16]. What is more, the government order of self-isolation forced people to spend almost all their time at home and limit or completely abandon human encounters. In addition, the deteriorating economic situation was the cause of financial hardship for many people. All these difficulties and challenges arose in the aura of a new, contagious disease with unexplored, long-lasting health effects and not fully known infectivity and lethality [17]. Dealing with the situation was not facilitated by the phenomenon of global misinformation, called by some experts as the "infodemic", which may be defined as an overabundance of information that makes it difficult for people to find trustworthy sources and reliable guidance [18]. Studies have shown that people have multiple ways of reacting to a crisis: from radical and even violent practices, towards individual solutions and depression [19]. Not only the challenges arising from the COVID-19 pandemic but also the ways of reacting to it and coping with it are issues of paramount importance that are worth investigating.

The reactions to unusual crisis situations may be dependent on dispositional factors, such as trait anxiety or perceived control [20, 21]. A study on reactions to Hurricane Hugo has shown that people with higher trait anxiety are more likely to develop posttraumatic symptoms following a natural disaster [20]. Moreover, lack of perceived control was shown to be positively related to the level of distress during an earthquake in Turkey [21]. According to some researchers, the COVID-19 crisis and natural disasters have much in common, as the emotions and behavior they cause are based on the same primal human emotion—fear [22]. Both pandemics and natural disasters disrupt people's everyday lives and may have severe economic, social and psychological consequences [23]. However, despite many similarities to natural disasters, COVID-19 is a unique situation—only in 2020, the current pandemic has taken more lives than the world's combined natural disasters in any of the past twenty years [24]. It needs to be noted that natural disasters may pose different challenges than health crises and for this reason, they may provoke disparate reactions [25]. Research on the reactions to former epidemics has shown that avoidance and safety behaviors, such as avoiding going out, visiting crowded places, and visiting hospitals, are widespread at such times [26]. When it comes to the ways of dealing with the current COVID-19 pandemic, a substantial part of the quantitative research on this issue focuses on coping mechanisms. Studies have shown that the most prevalent coping strategies are highly problem-focused [5]. Most people tend to listen to expert advice and behave calmly and appropriately in the face of the coronavirus outbreak [5]. Problem-focused coping is particularly characteristic of healthcare professionals. A study on Chinese nurses has shown that the closer the problem is to the person and the more fear it evokes, the more problem-focused coping strategy is used to deal with it [27]. On the other hand, a negative coping style that entails risky or aggressive behaviors, such as drug or alcohol use, is also used to deal with the challenges arising from the COVID-19 pandemic [28]. The factors that are correlated with negative coping include coronavirus anxiety, impairment, and suicidal ideation [28]. It is worth emphasizing that social support is a very important component of dealing with crises [29].

Scientists have attempted to systematize the reactions to difficult and unusual situations. One such concept is the "3 Cs" model created by Reich [30]. It accounts for the general rules of resilience in situations of stress caused by crises, such as natural disasters. The 3 Cs stand for:

control (a belief that personal resources can be accessed to achieve valued goals), coherence (the human desire to make meaning of the world), and connectedness (the need for human contact and support) [30]. Polizzi and colleagues [22] reviewed this model from the perspective of the current COVID-19 pandemic. The authors claim that natural disasters and COVID-19 pandemic have much in common and therefore, the principles of resilience in natural disaster situations can also be used in the situation of the current pandemic [22]. They propose a set of coping behaviors that could be useful in times of the coronavirus outbreak, which include control (e.g., planning activities for each day, getting adequate sleep, limiting exposure to the news, and helping others), coherence (e.g., mindfulness and developing a coherent narrative on the event), and connectedness (e.g., establishing new relationships and caring for existing social bonds) [22].

## Current study

The issue of the challenges arising from the current COVID-19 pandemic and the ways of coping with them is complex and many feelings accompanying these experiences may be unconscious and difficult to verbalize. Therefore, in order to explore and understand it deeply, qualitative methodology was applied. Although there were few qualitative studies on the reaction to the pandemic [e.g., 31–33], they did not capture the perception of the challenges and their changes that arise as the pandemic develops. Since the situation with the COVID-19 pandemic is very dynamic, the reactions to the various restrictions, orders or bans are evolving. Therefore, it was decided to conduct a qualitative longitudinal study with multiple interviews with the same respondents [34].

The study investigates the challenges arising from the current pandemic and the way people deal with them. The main aim of the project was to capture people's reactions to the unusual and unexpected situation of the COVID-19 pandemic. Therefore, the project was largely exploratory in nature. Interviews with the participants at different stages of the epidemic allowed us to see a wide spectrum of problems and ways of dealing with them. The conducted study had three main research questions:

- What are the biggest challenges connected to the COVID-19 pandemic and the resulting restrictions?

- How are people dealing with the pandemic challenges?

- What are the ways of coping with the restrictions resulting from a pandemic change as it continues and develops (perspective of first 6 months)?

## Methods

**Design.**   The study was approved by the institutional review board of the Faculty of Psychology University of Warsaw, Poland. All participants were provided written and oral information about the study, which included that participation was voluntary, that it was possible to withdraw without any consequences at any time, and the precautions that would be taken to protect data confidentiality. Informed consent was obtained from all participants. To ensure confidentiality, quotes are presented only with gender, age, and family status.

The study was based on qualitative methodology: individual in-depth interviews, s which are the appropriate to approach a new and unknown and multithreaded topic which, at the beginning of 2020, was the COVID-19 pandemic. Due to the need to observe respondents' reactions to the dynamically changing situation of the COVID-19 pandemic, longitudinal

study was used where the moderator met on-line with the same respondent several times, at specific time intervals. A longitudinal study was used to capture the changes in opinions, emotions, and behaviors of the respondents resulting from the changes in the external circumstances (qualitative in-depth interview tracking–[34]).

The study took place from the end of March to October 2020. Due to the epidemiological situation in the country interviews took place online, using the Google Meets online video platform. The audio was recorded and then transcribed. Before taking part in the project, the respondents were informed about the purpose of the study, its course, and the fact that participation in the project is voluntary, and that they will be able to withdraw from participation at any time. The respondents were not paid for taking part in the project.

**Participants.**   In total, 115 interviews were conducted with 20 participants (6 interviews with the majority of respondents). Two participants (number 11 and 19, S2 Table) dropped out of the last two interviews, and one (number 6) dropped out of the last interview. The study was based on a purposive sample and the respondents differed in gender, age, education, family status, and work situation (see S2 Table). In addition to demographic criteria intended to ensure that the sample was as diverse as possible, an additional criterion was to have a permanent Internet connection and a computer capable of online video interviewing. Study participants were recruited using the snowball method. They were distant acquaintances of acquaintances of individuals involved in the study. None of the moderators knew their interviewees personally.

A total of 10 men and 10 women participated in the study; their age range was: 25–55; the majority had higher education (17 respondents), they were people with different professions and work status, and different family status (singles, couples without children, and families with children). Such diversity of respondents allowed us to obtain information from different life perspectives. A full description of characteristics of study participants can be found in S2 Table.

**Procedure.**   Each interview took 2 hours on average, which gives around 240 hours of interviews. Subsequent interviews with the same respondents conducted at different intervals resulted from the dynamics of the development of the pandemic and the restrictions introduced in Poland by the government.

The interviews scenario took a semi-structured form. This allowed interviewers freely modify the questions and topics depending on the dynamics of the conversation and adapt the subject matter of the interviews not only to the research purposes but also to the needs of a given respondent. The interview guides were modified from week to week, taking into account the development of the epidemiological situation, while at the same time maintaining certain constant parts that were repeated in each interview. The main parts of the interview topic guide consisted of: (a) experiences from the time of previous interviews: thoughts, feeling, fears, and hopes; (b) everyday life—organization of the day, work, free time, shopping, and eating, etc.; (c) changes—what had changed in the life of the respondent from the time of the last interview; (d) ways of coping with the situation; and (e) media—reception of information appearing in the media. Additionally, in each interview there were specific parts, such as the reactions to the beginning of the pandemic in the first interview or the reaction to the specific restrictions that were introduced.

The interviews were conducted by 5 female interviewers with experience in moderating qualitative interviews, all with a psychological background. After each series of interviews, all the members of the research teams took part in debriefing sessions, which consisted of discussing the information obtained from each respondent, exchanging general conclusions, deciding about the topics for the following interview stage, and adjusting them to the pandemic situation in the country.

**Data analysis.**   All the interviews were transcribed in Polish by the moderators and then double-checked (each moderator transcribed the interviews of another moderator, and then the interviewer checked the accuracy of the transcription). The whole process of analysis was conducted on the material in Polish (the native language of the authors of the study and respondents). The final page count of the transcript is approximately 1800 pages of text. The results presented below are only a portion of the total data collected during the interviews. While there are about 250 pages of the transcription directly related to the topic of the article, due to the fact that the interview was partly free-form, some themes merge with others and it is not possible to determine the exact number of pages devoted exclusively to analysis related to the topic of the article. Full dataset can be found in S1 Dataset.

Data was then processed into thematic analysis, which is defined as a method of developing qualitative data consisting of the identification, analysis, and description of the thematic areas [35]. In this type of analysis, a thematic unit is treated as an element related to the research problem that includes an important aspect of data. An important advantage of thematic analysis is its flexibility, which allows for the adoption of the most appropriate research strategy to the phenomenon under analysis. An inductive approach was used to avoid conceptual tunnel vision. Extracting themes from the raw data using an inductive approach precludes the researcher from imposing a predetermined outcome.

As a first step, each moderator reviewed the transcripts of the interviews they had conducted. Each transcript was thematically coded individually from this point during the second and the third reading. In the next step, one of the researchers reviewed the codes extracted by the other members of the research team. Then she made initial interpretations by generating themes that captured the essence of the previously identified codes. The researcher created a list of common themes present in all of the interviews. In the next step, the extracted themes were discussed again with all the moderators conducting the coding in order to achieve consistency. This collaborative process was repeated several times during the analysis. Here, further superordinate (challenges of COVID-19 pandemic) and subordinate (ways of dealing with challenges) themes were created, often by collapsing others together, and each theme listed under a superordinate and subordinate category was checked to ensure they were accurately represented. Through this process of repeated analysis and discussion of emerging themes, it was possible to agree on the final themes that are described below.

## Results

**Main challenges of the COVID-19 pandemic.**   *Challenge 1 –limitation of direct contact with people.* The first major challenge of the pandemic was that direct contact with other people was significantly reduced. The lockdown forced many people to work from home and limit contact not only with friends but also with close family (parents, children, and siblings). Limiting contact with other people was a big challenge for most of our respondents, especially those who were living alone and for those who previously led an active social life. Depending on their earlier lifestyle profile, for some, the bigger problem was the limitation of contact with the family, for others with friends, and for still others with co-workers.

*I think that because I can't meet up with anyone and that I'm not in a relationship, I miss having sex, and I think it will become even more difficult because it will be increasingly hard to meet anyone. (5.3_ M_39_single).* The number In the brackets at the end of the quotes marks the respondent's number (according to Table 1) and the stage of the interview (after the dash), further is information about gender (F/M), age of the respondent and family status. Linguistic errors in the quotes reflect the spoken language of the respondents.

**Table 1. Challenges and ways to cope with them.**

| Challenge | Ways to cope |
|---|---|
| Social isolation | • Meetings on online platforms<br>• Circumventing the rules (e.g., exceptions for the family in the restriction of face-to-face meetings) |
| Restriction on movement and travel | • "Rewarding oneself", "indulging"–hedonic behaviors<br>• Making future travel plans |
| Necessary change in active lifestyle | • Looking for "substitutes", e.g., online exercises, using movie streaming platforms<br>• Dreaming about the future after the pandemic (e.g., travelling abroad) |
| Boredom, monotony | • Rituals<br>• Looking for new challenges and experiences (e.g., baking bread) |
| Uncertainty about the future | • Looking for information about the number of COVID-19 infections and deaths<br>• Cutting off from information |

***Changes over time***. Over the course of the 6 months of the study, an evolution in the attitudes to the restriction of face-to-face contact could be seen: from full acceptance, to later questioning its rationale. Initially (March and April), almost all the respondents understood the reasons for the isolation and were compliant. At the beginning, people were afraid of the unknown COVID-19. They were concerned that the tragic situation from Italy, which was intensively covered in the media, could repeat itself in Poland (stage 1–2 of the study). However, with time, the isolation started to bother them more and more, and they started to look for solutions to bypass the isolation guidelines (stage 3–4), both real (simply meeting each other) and mental (treating isolation only as a guideline and not as an order, perceiving the family as being less threatening than acquaintances or strangers in a store). The turning point was the long May weekend that, due to two public holidays (1st and 3rd May), has for many years been used as an opportunity to go away with family or friends. Many people broke their voluntary isolation during that time encouraged by information about the coming loosening of restrictions.

During the summer (stage 5 of the survey), practically no one was fully compliant with the isolation recommendations anymore. At that time, a growing familiarity could be observed with COVID-19 and an increasing tendency to talk about it as "one of many diseases", and to convince oneself that one is not at risk and that COVID-19 is no more threatening than other viruses. Only a small group of people consciously failed to comply with the restrictions of contact with others from the very beginning of the pandemic. This behavior was mostly observed among people who were generally less anxious and less afraid of COVID-19.

*I've had enough. I've had it with sitting at home. Okay, there's some kind of virus, it's as though it's out there somewhere; it's like I know 2 people who were infected but they're still alive, nothing bad has happened to anyone. It's just a tiny portion of people who are dying. And is it really such a tragedy that we have to be locked up at home? Surely there's an alternative agenda there? (17.4_F_35_Adult and child)*

***Ways of dealing***. In the initial phase, when almost everyone accepted this restriction and submitted to it, the use of communication platforms for social meetings increased (see Ways of dealing with challenges in Table 1). Meetings on communication platforms were seen as an

equivalent of the previous face-to-face contact and were often even accompanied by eating or drinking alcohol together. However, over time (at around stage 4–5 of the study) people began to feel that such contact was an insufficient substitute for face-to-face meetings and interest in online meetings began to wane. During this time, however, an interesting phenomenon could be seen, namely, that for many people the family was seen as a safer environment than friends, and definitely safer than strangers. The belief was that family members would be honest about being sick, while strangers not necessarily, and—on an unconscious level—the feeling was that the "family is safe", and the "family can't hurt them".

When it became clear that online communication is an insufficient substitute for face-to-face contacts, people started to meet up in real life. However, a change in many behaviors associated with meeting people is clearly visible, e.g.: refraining from shaking hands, refraining from cheek kissing to greet one another, and keeping a distance during a conversation.

*I can't really say that I could 'feel' Good Friday or Holy Saturday. On Sunday, we had breakfast together with my husband's family and his sister. We were in three different places but we connected over Skype. Later, at noon, we had some coffee with my parents, also over Skype. It's obvious though that this doesn't replace face-to-face contact but it's always some form of conversation. (9.3_F_25_Couple, no children)*

*Challenge 2 –restrictions on movement and travel.* In contrast to the restrictions on contact with other people, the restrictions on movement and the closing of borders were perceived more negatively and posed bigger challenges for some people (especially those who used to do a lot of travelling). In this case, it was less clear why these regulations were introduced (especially travel restrictions within the country). Moreover, travel restrictions, particularly in the case of international travels, were associated with a limitation of civil liberties. The limitation (or complete ban) on travelling abroad in the Polish situation evoked additional connotations with the communist times, that is, with the fact that there was no freedom of movement for Polish citizens (associations with totalitarianism and dictatorship). Interestingly, the lack of acceptance of this restriction was also manifested by people who did not travel much. Thus, it was not just a question of restricting travelling abroad but more of restricting the potential opportunity ("even if I'm not planning on going anywhere, I know I still can").

Limitations on travelling around the country were particularly negatively felt by families with children, where parents believe that regular exercise and outings are necessary for the proper development of their children. For parents, it was problematic to accept the prohibition of leaving the house and going to the playground (which remained closed until mid-May). Being outdoors was perceived as important for maintaining immunity (exercise as part of a healthy lifestyle), therefore, people could not understand the reason underlying this restriction and, as a consequence, often did not accept it.

*I was really bothered by the very awareness that I can't just jump in my car or get on a plane whenever I want and go wherever I want. It's not something that I have to do on a daily basis but freedom of movement and travelling are very important for me. (14.2_M_55_Two adults and children)*

**Changes over time**. The travel and movement limitations, although objectively less severe for most people, aroused much greater anger than the restrictions on social contact. This was probably due to a greater sense of misunderstanding as to why these rules were being introduced in the first place. Moreover, they were often communicated inconsistently and chaotically (e.g., a ban on entering forests was introduced while, at the same time, shopping malls

remained open and masses were allowed to attend church services). This anger grew over time —from interview to interview, the respondents' irritation and lack of acceptance of this was evident (culminating in the 3rd-4th stage of the study). The limitation of mobility was also often associated with negative consequences for both health and the economy. Many people are convinced that being in the open air (especially accompanied by physical activity) strengthens immunity, therefore, limiting such activity may have negative health consequences. Some respondents pointed out that restricting travelling, the use of hotels and restaurants, especially during the holiday season, will have serious consequences for the existence of the tourism industry.

> *I can't say I completely agree with these limitations because it's treating everything selectively. It's like the shopping mall is closed, I can't buy any shoes but I can go to a home improvement store and buy some wallpaper for myself. So I don't see the difference between encountering people in a home improvement store and a shopping mall. (18.2_F_48_Two adults and children)*

***Ways of dealing****.* Since the restriction of movement and travel was more often associated with pleasure-related behaviors than with activities necessary for living, the compensations for these restrictions were usually also from the area of hedonistic behaviors. In the statements of our respondents, terms such as "indulging" or "rewarding oneself" appeared, and behaviors such as throwing small parties at home, buying better alcohol, sweets, and new clothes were observed. There were also increased shopping behaviors related to hobbies (sometimes hobbies that could not be pursued at the given time)–a kind of "post-pandemic" shopping spree (e.g., a new bike or new skis).

Again, the reaction to this restriction also depended on the level of fear of the COVID-19 disease. People who were more afraid of being infected accepted these restrictions more easily as it gave them the feeling that they were doing something constructive to protect themselves from the infection. Conversely, people with less fears and concerns were more likely to rebel and break these bans and guidelines.

Another way of dealing with this challenge was making plans for interesting travel destinations for the post-pandemic period. This was especially salient in respondents with an active lifestyle in the past and especially visible during the 5th stage of the study.

> *Today was the first day when I went to the store (due to being in quarantine after returning from abroad). I spent loads of money but I normally would have never spent so much on myself. I bought sweets and confectionery for Easter time, some Easter chocolates, too. I thought I'd do some more baking so I also bought some ingredients to do this. (1.2_F_25_single)*

*Challenge 3 –necessary change in active lifestyle.* Many of the limitations related to COVID-19 were a challenge for people with an active lifestyle who would regularly go to the cinema, theater, and gym, use restaurants, and do a lot of travelling. For those people, the time of the COVID constraints has brought about huge changes in their lifestyle. Most of their activities were drastically restricted overnight and they suddenly became domesticated by force, especially when it was additionally accompanied by a transition to remote work.

Compulsory spending time at home also had serious consequences for people with school-aged children who had to confront themselves with the distance learning situation of their children. The second challenge for families with children was also finding (or helping find) activities for their children to do in their free time without leaving the house.

*I would love to go to a restaurant somewhere. We order food from the restaurant at least once a week, but I'd love to go to the restaurant. Spending time there is a different way of functioning. It is enjoyable and that is what I miss. I would also go to the cinema, to the theater. (13.3_M_46_Two adults and child.)*

**Changes over time**. The nuisance of restrictions connected to an active lifestyle depended on the level of restrictions in place at a given time and the extent to which a given activity could be replaced by an alternative. Moreover, the response to these restrictions depended more on the individual differences in lifestyle rather than on the stage of the interview (except for the very beginning, when the changes in lifestyle and everyday activities were very sudden).

*I miss that these restaurants are not open. And it's not even that I would like to eat something specific. It is in all of this that I miss such freedom the most. It bothers me that I have no freedom. And I am able to get used to it, I can cook at home, I can order from home. But I just wish I had a choice. (2.6_F_27_single).*

**Ways of dealing**. In the initial phase of the pandemic (March-April—stage 1–3 of the study), when most people were afraid of the coronavirus, the acceptance of the restrictions was high. At the same time, efforts were made to find activities that could replace existing ones. Going to the gym was replaced by online exercise, and going to the cinema or theater by intensive use of streaming platforms. In the subsequent stages of the study, however, the respondents' fatigue with these "substitutes" was noticeable. It was then that more irritation and greater non-acceptance of certain restrictions began to appear. On the other hand, the changes or restrictions introduced during the later stages of the pandemic were less sudden than the initial ones, so they were often easier to get used to.

*I bought a small bike and even before that we ordered some resistance bands to work out at home, which replace certain gym equipment and devices. [. . .] I'm considering learning a language. From the other online things, my girlfriend is having yoga classes, for instance. (7.2_M_28_Couple, no children)*

*Challenge 4 –boredom, monotony*. As has already been shown, for many people, the beginning of the pandemic was a huge change in lifestyle, an absence of activities, and a resulting slowdown. It was sometimes associated with a feeling of weariness, monotony, and even of boredom, especially for people who worked remotely, whose days began to be similar to each other and whose working time merged with free time, weekdays with the weekends, and free time could not be filled with previous activities.

*In some way, boredom. I can't concentrate on what I'm reading. I'm trying to motivate myself to do such things as learning a language because I have so much time on my hands, or to do exercises. I don't have this balance that I'm actually doing something for myself, like reading, working out, but also that I'm meeting up with friends. This balance has gone, so I've started to get bored with many things. Yesterday I felt that I was bored and something should start happening. (. . .) After some time, this lack of events and meetings leads to such immense boredom. (1.5_F_25_single)*

**Changes over time**. The feeling of monotony and boredom was especially visible in stage 1 and 2 of the study when the lockdown was most restrictive and people were knocked out of their daily routines. As the pandemic continued, boredom was often replaced by irritation in

some, and by stagnation in others (visible in stages 3 and 4 of the study) while, at the same time, enthusiasm for taking up new activities was waning. As most people were realizing that the pandemic was not going to end any time soon, a gradual adaptation to the new lifestyle (slower and less active) and the special pandemic demands (especially seen in stage 5 and 6 of the study) could be observed.

> *But I see that people around me, in fact, both family and friends, are slowly beginning to prepare themselves for more frequent stays at home. So actually more remote work, maybe everything will not be closed and we will not be locked in four walls, but this tendency towards isolation or self-isolation, such a deliberate one, appears. I guess we are used to the fact that it has to be this way. (15.6_M_43_Two adults and child).*

***Ways of dealing***. The answer to the monotony of everyday life and to finding different ways of separating work from free time was to stick to certain rituals, such as "getting dressed for work", even when work was only by a computer at home or, if possible, setting a fixed meal time when the whole family would gather together. For some, the time of the beginning of the pandemic was treated as an extra vacation. This was especially true of people who could not carry out their work during the time of the most severe restrictions (e.g., hairdressers and doctors). For them, provided that they believed that everything would return to normal and that they would soon go back to work, a "vacation mode" was activated wherein they would sleep longer, watch a lot of movies, read books, and generally do pleasant things for which they previously had no time and which they could now enjoy without feeling guilty. Another way of dealing with the monotony and transition to a slower lifestyle was taking up various activities for which there was no time before, such as baking bread at home and cooking fancy dishes.

> *I generally do have a set schedule. I begin work at eight. Well, and what's changed is that I can get up last minute, switch the computer on and be practically making my breakfast and coffee during this time. I do some work and then print out some materials for my younger daughter. You know, I have work till four, I keep on going up to the computer and checking my emails. (19.1_F_39_Two adults and children)*

***Challenge 5 –uncertainty about the future***. Despite the difficulties arising from the circumstances and limitations described above, it seems that psychologically, the greatest challenge during a pandemic is the uncertainty of what will happen next. There was a lot of contradictory information in the media that caused a sense of confusion and heightened the feeling of anxiety.

> *I'm less bothered about the changes that were put in place and more about this concern about what will happen in the future. Right now, it's like there's these mood swings. [. . .] Based on what's going on, this will somehow affect every one of us. And that's what I'm afraid of. The fact that someone will not survive and I have no way of knowing who this could be—whether it will be me or anyone else, or my dad, if somehow the coronavirus will sneak its way into our home. I simply don't know. I'm simply afraid of this. (10.1_F_55_Couple, no children)*

***Changes over time***. In the first phase of the pandemic (interviews 1–3), most people felt a strong sense of not being in control of the situation and of their own lives. Not only did the consequences of the pandemic include a change in lifestyle but also, very often, the suspension of plans altogether. In addition, many people felt a strong fear of the future, about what would happen, and even a sense of threat to their own or their loved ones' lives. Gradually (interview

4), alongside anxiety, anger began to emerge about not knowing what would happen next. At the beginning of the summer (stage 5 of the study), most people had a hope of the pandemic soon ending. It was a period of easing restrictions and of opening up the economy. Life was starting to look more and more like it did before the pandemic, fleetingly giving an illusion that the end of the pandemic was "in sight" and the vision of a return to normal life. Unfortunately, autumn showed that more waves of the pandemic were approaching. In the interviews of the 6[th] stage of the study, we could see more and more confusion and uncertainty, a loss of hope, and often a manifestation of disagreement with the restrictions that were introduced.

> *This is making me sad and angry. More angry, in fact. [. . .] I don't know what I should do. Up until now, there was nothing like this. Up until now, I was pretty certain of what I was doing in all the decisions I was making. (14.4_M_55_Two adults and children)*

***Ways of dealing***. People reacted differently to the described feeling of insecurity. In order to reduce the emerging fears, some people searched (sometimes even compulsively) for any information that could help them "take control" of the situation. These people searched various sources, for example, information on the number of infected persons and the number of deaths. This knowledge gave them the illusion of control and helped them to somewhat reduce the anxiety evoked by the pandemic. The behavior of this group was often accompanied by very strict adherence to all guidelines and restrictions (e.g., frequent hand sanitization, wearing a face mask, and avoiding contact with others). This behavior increased the sense of control over the situation in these people.

A completely opposite strategy to reducing the feeling of uncertainty which we also observed in some respondents was cutting off information in the media about the scale of the disease and the resulting restrictions. These people, unable to keep up with the changing information and often inconsistent messages, in order to maintain cognitive coherence tried to cut off the media as much as possible, assuming that even if something really significant had happened, they would still find out.

> *I want to keep up to date with the current affairs. Even if it is an hour a day. How is the pandemic situation developing—is it increasing or decreasing. There's a bit of propaganda there because I know that when they're saying that they have the situation under control, they can't control it anyway. Anyhow, it still has a somewhat calming effect that it's dying down over here and that things aren't that bad. And, apart from this, I listen to the news concerning restrictions, what we can and can't do. (3.1_F_54_single)*

## Discussion and conclusions

### Discussion

The results of our study showed that the five greatest challenges resulting from the COVID-19 pandemic are: limitations of direct contact with people, restrictions on movement and travel, change in active lifestyle, boredom and monotony, and finally uncertainty about the future. As we can see the spectrum of problems resulting from the pandemic is very wide and some of them have an impact on everyday functioning and lifestyle, some other influence psychological functioning and well-being. Moreover, different people deal with these problems differently and different changes in everyday life are challenging for them. The first challenge of the pandemic COVID-19 problem is the consequence of the limitation of direct contact with others.

This regulation has very strong psychological consequences in the sense of loneliness and lack of closeness. Initially, people tried to deal with this limitation through the use of internet communicators. It turned out, however, that this form of contact for the majority of people was definitely insufficient and feelings of deprivation quickly increased. As much data from psychological literature shows, contact with others can have great psychological healing properties [e.g., 29]. The need for closeness is a natural need in times of crisis and catastrophes [30]. Unfortunately, during the COVID-19 pandemic, the ability to meet this need was severely limited by regulations. This led to many people having serious problems with maintaining a good psychological condition.

Another troubling limitation found in our study were the restrictions on movement and travel, and the associated restrictions of most activities, which caused a huge change in lifestyle for many people. As shown in previous studies, travel and diverse leisure activities are important predictors of greater well-being [36]. Moreover, COVID-19 pandemic movement restrictions may be perceived by some people as a threat to human rights [37], which can contribute to people's reluctance to accept lockdown rules.

The problem with accepting these restrictions was also related to the lack of understanding of the reasons behind them. Just as the limitation in contact with other people seemed understandable, the limitations related to physical activity and mobility were less so. Because of these limitations many people lost a sense of understanding of the rules and restrictions being imposed. Inconsistent communication in the media—called by some researchers the 'infodemic' [18], as well as discordant recommendations in different countries, causing an increasing sense of confusion in people.

Another huge challenge posed by the current pandemic is the feeling of uncertainty about the future. This feeling is caused by constant changes in the rules concerning daily functioning during the pandemic and what is prohibited and what is allowed. People lose their sense of being in control of the situation. From the psychological point of view, a long-lasting experience of lack of control can cause so-called learned helplessness, a permanent feeling of having no influence over the situation and no possibility of changing it [38], which can even result in depression and lower mental and physical wellbeing [39]. Control over live and the feeling that people have an influence on what happens in their lives is one of the basic rules of crisis situation resilience [30]. Unfortunately, also in this area, people have huge deficits caused by the pandemic. The obtained results are coherent with previous studies regarding the strategies harnessed to cope with the pandemic [e.g., 5, 10, 28, 33]. For example, some studies showed that seeking social support is one of the most common strategies used to deal with the coronavirus pandemic [33, 40]. Other ways to deal with this situation include distraction, active coping, and a positive appraisal of the situation [41]. Furthermore, research has shown that simple coping behaviors such as a healthy diet, not reading too much COVID-19 news, following a daily routine, and spending time outdoors may be protective factors against anxiety and depressive symptoms in times of the coronavirus pandemic [41].

This study showed that the acceptance of various limitations, and especially the feeling of discomfort associated with them, depended on the person's earlier lifestyle. The more active and socializing a person was, the more restrictions were burdensome for him/her. The second factor, more of a psychological nature, was the fear of developing COVID-19. In this case, people who were more afraid of getting sick were more likely to submit to the imposed restrictions that, paradoxically, did not reduce their anxiety, and sometimes even heightened it.

**Limitations of the study.** While the study shows interesting results, it also has some limitations. The purpose of the study was primarily to capture the first response to problems resulting from a pandemic, and as such its design is not ideal. First, the study participants are not diverse as much as would be desirable. They are mostly college-educated and relatively

well off, which may influence how they perceive the pandemic situation. Furthermore, the recruitment was done by searching among the further acquaintances of the people involved in the study, so there is a risk that all the people interviewed come from a similar background. It would be necessary to conduct a study that also describes the reaction of people who are already in a more difficult life situation before the pandemic starts.

Moreover, it would also be worthwhile to pay attention to the interviewers themselves. All of the moderators were female, and although gender effects on the quality of the interviews and differences between the establishment of relationships between women and men were not observed during the debriefing process, the topic of gender effects on the results of qualitative research is frequently addressed in the literature [42, 43]. Although the researchers approached the process with reflexivity and self-criticism at all stages, it would have seemed important to involve male moderators in the study to capture any differences in relationship dynamics.

**Practical implications.** The study presented has many practical implications. Decision-makers in the state can analyze the COVID-19 pandemic crisis in a way that avoids a critical situation involving other infectious diseases in the future. The results of our study showing the most disruptive effects of the pandemic on people can serve as a basis for developing strategies to deal with the effects of the crisis so that it does not translate into a deterioration of the public's mental health in the future.

The results of our study can also provide guidance on how to communicate information about restrictions in the future so that they are accepted and respected (for example by giving rational explanations of the reasons for introducing particular restrictions). In addition, the results of our study can also be a source of guidance on how to deal with the limitations that may arise in a recurrent COVID-19 pandemic, as well as other emergencies that could come.

The analysis of the results showed that the COVID-19 pandemic, and especially the lockdown periods, are a particular challenge for many people due to reduced social contact. On the other hand, it is social contacts that are at the same time a way of a smoother transition of crises. This knowledge should prompt decision-makers to devise ways to ensure pandemic safety without drastically limiting social contacts and to create solutions that give people a sense of control (instead of depriving it of). Providing such solutions can reduce the psychological problems associated with a pandemic and help people to cope better with it.

## Conclusions

As more and more is said about the fact that the COVID-19 pandemic may not end soon and that we are likely to face more waves of this disease and related lockdowns, it is very important to understand how the different restrictions are perceived, what difficulties they cause and what are the biggest challenges resulting from them. For example, an important element of accepting the restrictions is understanding their sources, i.e., what they result from, what they are supposed to prevent, and what consequences they have for the fight against the pandemic. Moreover, we observed that the more incomprehensible the order was, the more it provoked to break it. This means that not only medical treatment is extremely important in an effective fight against a pandemic, but also appropriate communication.

The results of our study showed also that certain restrictions cause emotional deficits (e.g., loneliness, loss of sense of control) and, consequently, may cause serious problems with psychological functioning. From this perspective, it seems extremely important to understand which restrictions are causing emotional problems and how they can be dealt with in order to reduce the psychological discomfort associated with them.

## Supporting information

**S1 Table. A full description of the changes occurring in Poland at the time of the study.**
(DOCX)

**S2 Table. Characteristics of study participants.**
(DOCX)

**S1 Dataset. Transcriptions from the interviews.**
(ZIP)

## Author Contributions

**Conceptualization:** Dominika Maison, Dominika Adamczyk, Daria Affeltowicz.

**Formal analysis:** Diana Jaworska, Dominika Adamczyk, Daria Affeltowicz.

**Funding acquisition:** Dominika Maison.

**Investigation:** Diana Jaworska, Dominika Adamczyk, Daria Affeltowicz.

**Methodology:** Dominika Maison, Dominika Adamczyk, Daria Affeltowicz.

**Project administration:** Dominika Maison.

**Supervision:** Dominika Maison.

**Writing – original draft:** Dominika Maison, Diana Jaworska, Dominika Adamczyk.

**Writing – review & editing:** Dominika Maison, Diana Jaworska, Dominika Adamczyk.

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
