## [Decision Letter · Decision Letter 0]

17 May 2021

PONE-D-21-11248

The challenges arising from the COVID-19 pandemic and the way people deal with them. A qualitative longitudinal study.

PLOS ONE

Dear Dr. Dominika,

Thank you for submitting your manuscript to PLOS ONE. After careful consideration, we feel that it has merit but does not fully meet PLOS ONE’s publication criteria as it currently stands. Therefore, we invite you to submit a revised version of the manuscript that addresses the points raised during the review process.

We look forward to receiving your revised manuscript.

Kind regards,

Shah Md Atiqul Haq

Academic Editor

PLOS ONE

Journal Requirements:

2. In order to improve reporting, in your methods section, please provide additional information about the participant recruitment method, including a description of any inclusion/exclusion criteria that were applied to participant recruitment and a description of how participants were recruited.

3.We note that you have indicated that data from this study are available upon request. PLOS only allows data to be available upon request if there are legal or ethical restrictions on sharing data publicly. For information on unacceptable data access restrictions, please see http://journals.plos.org/plosone/s/data-availability#loc-unacceptable-data-access-restrictions.

4. Please include a copy of Table 3 which you refer to in your text on page 32.

Additional Editor Comments:

Dear authors/writers.

I would like to ask you to revise the article as suggested by the commenters.

You also need to focus on the discussion section. This section should be logical, concise, and have a strong conclusion.

Good luck!

Reviewers' comments:

Reviewer's Responses to Questions

**Comments to the Author**

1. Is the manuscript technically sound, and do the data support the conclusions?

Reviewer #1: Yes

Reviewer #2: No

Reviewer #3: Partly

2. Has the statistical analysis been performed appropriately and rigorously? 

Reviewer #1: Yes

Reviewer #2: No

Reviewer #3: No

3. Have the authors made all data underlying the findings in their manuscript fully available?

Reviewer #1: No

Reviewer #2: No

Reviewer #3: Yes

4. Is the manuscript presented in an intelligible fashion and written in standard English?

Reviewer #1: Yes

Reviewer #2: No

Reviewer #3: Yes

5. Review Comments to the Author

Reviewer #1: GENERAL COMMENTS

This manuscript analyses in depth the reactions of a Polish sample during the COVID pandemic. Its main strength is the longitudinal perspective qualitative study. However, the present manuscript fails at research structure and it is recommended to relay into its main contribution.

I have suggested some changes or recommendation which I hope the authors find them in a constructive manner.

My recommendation for this article is a major revision.

SPECIFIC COMMENTS

INTRODUCTION

Please add a reference for making those statements (L. 34, 51,54, 131, 136)

L 43 This study is not the first with a qualitative methodology, please revise the whole manuscript.

L 54 which reactions are going to be reported? reactions in people, policies, government ¿?

L 66-88 Authors describe the 6 stages and the research questions could fit better at the procedure subsection in the methods section. I would suggest describing the evolution of the COVID19 different stages in Poland should be detailed as the reader could understand better the study.

L 99 include the name of the main author of the 3C model

Still in this part, I would suggest authors linking the natural disaster situation and this health pandemic situation.

L 186 please provide the first author of this model

L 199-201 This sentence is no needed:

"However, it needs to be noted that Polizzi and colleagues (12) did not validate the

model in an empirical study but based their recommendations on theoretical considerations and conclusions from previous research."

L 209-211- It seems that some other qualitative studies have been done on COVID 19. Then please review the writing of the whole manuscript as it may not be the first qualitative study. Although, authors could point out their main contribution comparing their study with the previous study: it is a longitudinal study.

L 224-226 The aims or research questions should be in the same line as the results structure:

A) COVID 19 Challenges and how to cope with them over time.

B) How the psychological principles for natural disasters (3c's) could be a reference model to analyse the COVID 19 reactions?

The authors do not mention here in these research questions the 3C's models which should be taken into consideration.

L 219 an "s" is missing previous to reactions.

METHODS

L 228 Please omit the word research and insert Methods

The divide the content into these subsections:

Design

Participants

Tools

Procedure

Data analysis

L 245 google meets is an online video platform not for chat (writing). The authors should clarify this point.

L 246 it is strange to say "use a video camera" when the computers have it already and in google meet you only have to "open" it.

L 254 Please take into account this matter in the limitations sections as those with less than higher education may be in a worse financial situation so this study may be biased in that way.

The sample is 20 participants which were interviewed 6 times.

Table 1- Please explain at the bottom the meaning of

DINKS

Table 1- Please make a support sentence of why dividing the sample into more or less than 40 years old. One participant with 40 is young or old?

L 276 up to here there were in-depth- interviews…now you write here semi-structured interviews. Please define which was taken into consideration.

L 300-305- after the procedure the data analysis subsection should be mentioned. Could please give us some information about:

It is an inductive, deductive or mixed approach.

• How many pages were analyzed from the transcription?

• How many units of content did you find?

• Did you divide them into dimensions after or previously? To facilitate the analysis. Which criteria were taken?

• Did you find supra or subordinate factors in the analysis?

How did you find an agreement about the units of contents between the 5 researchers?

L 580 please check the spelling as "wanin" is not a word.

Table 2. Could this table be organized with the units of content found in each challenge and ways to cope?

L 821 Should state Table 3 now is Table 2 written.

L 693 This model has been explained before in the introduction please do not repeat the information in the results section.

The question is why this model could be adequate for analyzing the reactions during health pandemic situation?

DISCUSSION

L 824 Please separate Discussion and conclusions

Within the discussion authors should:

First paragraph: relate the main results with the aims-research questions of the study.

Second: detail and compare each result with previous results

Third: add the limitations of the study

Fourth: add some practical implications of the study

Fith: the final paragraph should be the conclusions.

Right now the discussion is too long in terms of length and the authors within 3 pages only have 8 references.

It could be shortened and more focused on the contribution of the paper.

Reviewer #2: (1) Is the manuscript technically sound, and do the data support the conclusions? [NO]

Three main problems:

First, the authors do not describe how the subjects were recruited, aside from saying it was a purposive sample. This is inadequate. Were there prior relationships between the subjects and the investigators?

Second, more needs to be said about the development of the data themes. The five challenges surely do not capture all the richness of 120 interviews. It would be helpful to hear about variability within the major themes as well as about what minor themes were identified.

Third, the conclusions go beyond the data presented in the results. Statements such as (line 719) “The analysis of the results showed that the first solution (full submission to orders) occurred more often in people with higher COVID-19 anxiety.” are unsupported by data presented in the results. Worse, we are not even told how anxiety was measured. The conclusion is peppered with unsupported statements of this sort, which are testable, but no results have been shown.

I recommend a table or tables showing the emergence of the major themes (concerns) across time and across subject characteristics. This would provide a framework for presenting and testing statements such as the one identified above.

(2) Has the statistical analysis been performed appropriately and rigorously? [NO]

There are appropriate qualitative statistics for testing most, if not all, of the conclusions the authors wish to highlight.

(3) Have the authors made all data underlying the findings in their manuscript fully available? [NO]

Generally, interview data can be anonymized by appropriate redaction of names and other sensitive material. If this is not possible, the code sheets showing the extraction of themes from the interviews would be the next best thing.

(4) Is the manuscript presented in an intelligible fashion and written in standard English? [NO]

There are some problems of English grammar, although these are not any worse than many late drafts by authors whose first language is English.

• E.g.: lines 101, 104-107, 219, 634.

• Table 1. Use of undefined acronym (DINKS).

Reviewer #3: This is a longitudinal, qualitative study on the challenges arising from the COVID-19 pandemic and the way people deal with them. As the pandemic is still ongoing, the impact may yet to be concluded in the background. This is a long article and many parts can be condensed and tightened up. For instance, the description about interviews on page 14 is redundant. Some other themes also seem redundant in some ways. Ways of dealing and change of lifestyle may be considered to be combined.

The authors claimed that none of the studies have employed qualitative methods, but is there any literature of quantitative study on similar topics, as this may serve as a baseline for this study.

There are also some concerns about methodology and results as follows:

Methods:

1. The authors interviewed 20 participants, six times each, and totaling 120 interviews. The interviewees range from 25-55. Why were participants divided into two age cohorts on the basis of age 40? There were 6 out of 20 interviewers who are over 40. The dividing of the age groups needs clearer reasoning. Particularly, most of citations used as evidence are based on groups under 40 years old.

2. The Interviews were conducted online by 5 female interviewers with experience in Psychology. Does the gender of the interviewers make any difference? The educational background and occupations of the interviewees were included in the demographics, but how this factor may affects the results is not explained.

3. All the interviews were transcribed, but there is no description about the ways for ensuring data accuracy and translation. (I suppose the interview was conducted in Polish?)

Results:

1. The authors used quotations of the interviewees to support the thematic points. Some point has 2 or 3 citations, some used only one. As this is already a fairly long article, one more representative citation for each point might be enough? Or, the responses can be integrated into the description of the findings using quotation.

2. Most of citations used as evidence are under aged 40 groups. This requires some further discussion and explanation.

3. The authors may consider restructure the results section and tighten up some findings.

4. Do different age, occupation, lifestyle, even occupation have any impact on how the interviewees experience different restriction.

5. The limitation of this study needs to be described, such as the representativeness of the respondents

Compared with the long presentation of results, the discussion is quite brief. Some parts require further explanation. For instance, p.40 “the study allowed deep understanding to be gained of the difficulties….” is not clear.

The paper ends abruptly. It requires a conclusion section to summarize the findings and its contribution/ or significance.

6. PLOS authors have the option to publish the peer review history of their article (what does this mean?). If published, this will include your full peer review and any attached files.

Reviewer #1: **Yes: **Cristina López de Subijana

Reviewer #2: No

Reviewer #3: No

---

## [Author Response · Author response to Decision Letter 0]

12 Jul 2021

Dear Reviewers,

Thank you for your valuable time and useful contribution to our article titled “The challenges arising from the COVID-19 pandemic and the way people deal with them. A qualitative longitudinal study”. We appreciate the inputs you have given, and we are sure that it will definitely help improve our manuscript.

Please see the "Response to Reviewers" file for detailed information on the changes made and responses to all comments. In general, we have made significant changes to the paper presented. First, we significantly restructured the method description section. We provided additional information about the participant recruitment method, including a description of inclusion criteria that were applied to participant recruitment and a description of how participants were recruited. As suggested, we have described the study procedure and the data analysis process itself in more detail. We have also shortened the description of the results, not limiting the description of the conclusions themselves, but reducing the number of quotations illustrating the obtained data. The discussions and conclusions section has been rewritten - according to your suggestions it includes the main results of the study, their comparison with previous results, limitations of the study, some practical implications of the study. Moreover, the whole article was corrected in terms of language and formatting (correcting errors in table names). 

We would like to thank you once again for all your comments and suggestions for changes to our article. We hope that after the changes we have made, the manuscript meets PLOS ONE standards, requirements, and publication criteria, and will appeal to the journal’s readers.

---

## [Decision Letter · Decision Letter 1]

26 Jul 2021

PONE-D-21-11248R1

The challenges arising from the COVID-19 pandemic and the way people deal with them. A qualitative longitudinal study.

PLOS ONE

Dear Dominika Adamczyk,

Thank you for submitting your manuscript to PLOS ONE. After careful consideration, we feel that it has merit but does not fully meet PLOS ONE’s publication criteria as it currently stands. Therefore, we invite you to submit a revised version of the manuscript that addresses the points raised during the review process.

We look forward to receiving your revised manuscript.

Kind regards,

Shah Md Atiqul Haq

Academic Editor

PLOS ONE

Journal Requirements:

Additional Editor Comments:

Dear author,

Please address the comments and suggestions asked by the reviewers.

Best of luck

Reviewers' comments:

Reviewer's Responses to Questions

**Comments to the Author**

1. If the authors have adequately addressed your comments raised in a previous round of review and you feel that this manuscript is now acceptable for publication, you may indicate that here to bypass the “Comments to the Author” section, enter your conflict of interest statement in the “Confidential to Editor” section, and submit your "Accept" recommendation.

Reviewer #1: All comments have been addressed

Reviewer #2: All comments have been addressed

2. Is the manuscript technically sound, and do the data support the conclusions?

Reviewer #1: Yes

Reviewer #2: Partly

3. Has the statistical analysis been performed appropriately and rigorously? 

Reviewer #1: Yes

Reviewer #2: N/A

4. Have the authors made all data underlying the findings in their manuscript fully available?

Reviewer #1: Yes

Reviewer #2: Yes

5. Is the manuscript presented in an intelligible fashion and written in standard English?

Reviewer #1: Yes

Reviewer #2: Yes

6. Review Comments to the Author

Reviewer #1: Dear Authors,

The manuscript has improved in the way now it is better supported and structured. Still I would suggest authors to shorten the length of the article. Almost fifty pages is too much for a research article. I would suggest to review each part and try to shorten some pages in each section. For example introduction is 5 pages and literature review another 5 pages. Research articles are normally around 3 pages for the whole introduction.

Reviewer #2: From a snowball sample of 20 respondents, the authors collected 115 interviews in six waves between March and October 2020 regarding perceptions of and responses to the COVID-19 pandemic. Themes were extracted from the interviews and organized into five major challenges that the respondents faced during this period. These challenges were found to be consistent with John Reich’s “3-Cs” and theory of resilience in natural disasters.

Generally, this is an interesting, opportunistic use of the pandemic. The topic is not insignificant. The snowball technique of sample development, while it has flaws, is an accepted method for small scale, investigative studies. The interview and data collection methodology, as described, appears to be generally rigorous. The issue of the uniform gender of the interviewers seems to me, given the small sample size, to be less of an issue than the relatively large number of interviewers for the sample. Fortunately, the study includes no sub-group analysis, so these issues become largely irrelevant. There is, inevitably, some danger that the analytic framework espoused by the authors influenced the data collection process, but, given the exploratory nature of the study, this is not a large concern.

The introduction, literature review, motivation for the study, methods, description of the participants, and description of the analytic process are all well done. Table 1 would be more informative if it showed the distribution of subjects over demographic characteristics, rather than listing each subject.

The reporting of the results I find less satisfactory. The methodology describes extracting themes from the interview data. The results report on challenges. Are the five challenges the only themes that were extracted? Are the challenges composed of clusters of themes? Do the challenges capture all the themes identified? By focusing on challenges, are the authors not talking about successful coping behaviors? For example, the subjects’ reactions to the limitations on travel are uniformly portrayed as negative (limitations on freedom, etc.) did none of the subjects perceive this as an appropriate public health measure that gave the subjects some control over the situation? A table showing the count of themes occurrences by study period would be greatly appreciated.

The interpretation of the results in the context of Reich’s framework is totally baffling in that it does not appear to refer to the results at all. There is one mention of one of the challenges in this section. Table 3 refers to some things that might be themes, but without a proper reporting of the themes, we don’t really know and are unable to judge its completeness. Without a better tie to the data, this section should be deleted.

The discussion and conclusion are generally adequate, given my concerns above. The limitations are accurately stated. My greatest concern here is that the process of focusing on concerns has caused the authors to lose sight of successful coping behaviors.

7. PLOS authors have the option to publish the peer review history of their article (what does this mean?). If published, this will include your full peer review and any attached files.

Reviewer #1: **Yes: **Cristina López de Subijana

Reviewer #2: No

---

## [Author Response · Author response to Decision Letter 1]

31 Aug 2021

Dear Editor and Reviewers,

Thank you once again for your valuable time and useful contribution to our article titled “The challenges arising from the COVID-19 pandemic and the way people deal with them. A qualitative longitudinal study”. The two most important changes we have made are the removal of the part of the description of the results presenting them in the light of the 3C theory and the significant shortening of the whole article. Please find our answers to the reviewers’ comments below. 

Reviewer #1: 

• Still I would suggest authors to shorten the length of the article. Almost fifty pages is too much for a research article. I would suggest to review each part and try to shorten some pages in each section. For example introduction is 5 pages and literature review another 5 pages. Research articles are normally around 3 pages for the whole introduction.

- This is the main change we have made in the revised article. We have significantly shortened the text, as suggested by both reviewers. 

Reviewer #2: 

• Table 1 would be more informative if it showed the distribution of subjects over demographic characteristics, rather than listing each subject.

- In our view, presenting the demographic characteristics of each respondent adds value to the text and shows the diversity of the respondents. However, in line with previous suggestions regarding the length of the text, we have decided to move the table to an appendix. 

• The reporting of the results I find less satisfactory. The methodology describes extracting themes from the interview data. The results report on challenges. Are the five challenges the only themes that were extracted? Are the challenges composed of clusters of themes? Do the challenges capture all the themes identified? 

- As presented in the text, the manuscript contains only part of the results, as it would be impossible to present all the findings from such a large number of interviews. For example, we do not describe specific changes in emotions here, and some residual information on this topic was removed from the text after the previous round of reviews because it would have been misleading throughout the text. 

• By focusing on challenges, are the authors not talking about successful coping behaviors?

• For example, the subjects’ reactions to the limitations on travel are uniformly portrayed as negative (limitations on freedom, etc.) did none of the subjects perceive this as an appropriate public health measure that gave the subjects some control over the situation?

• My greatest concern here is that the process of focusing on concerns has caused the authors to lose sight of successful coping behaviors.

- We understand where these doubts come from, but we cannot write about the positive aspects of the changes described, as they did not appear in our study. The participants perceived these limitations in an unambiguously negative way - perhaps a study conducted over a different period or one in which the participants retrospectively evaluate certain changes would have provided different results. However, in this paper, we describe the experience at the time of the study. 

• The interpretation of the results in the context of Reich’s framework is totally baffling in that it does not appear to refer to the results at all. There is one mention of one of the challenges in this section. Table 3 refers to some things that might be themes, but without a proper reporting of the themes, we don’t really know and are unable to judge its completeness. Without a better tie to the data, this section should be deleted.

- After careful consideration, we decided to comply with the above comment and remove the section presenting the results in the light of Reich's theory.

---

## [Decision Letter · Decision Letter 2]

20 Sep 2021

The challenges arising from the COVID-19 pandemic and the way people deal with them. A qualitative longitudinal study.

PONE-D-21-11248R2

Dear Dr. Dominika Adamczyk,

We’re pleased to inform you that your manuscript has been judged scientifically suitable for publication and will be formally accepted for publication once it meets all outstanding technical requirements.

Kind regards,

Shah Md Atiqul Haq

Academic Editor

PLOS ONE

Additional Editor Comments (optional):

Dear authors,

Thank you.

The article is now accepted.

Best wishes,

Reviewers' comments:

Reviewer's Responses to Questions

**Comments to the Author**

1. If the authors have adequately addressed your comments raised in a previous round of review and you feel that this manuscript is now acceptable for publication, you may indicate that here to bypass the “Comments to the Author” section, enter your conflict of interest statement in the “Confidential to Editor” section, and submit your "Accept" recommendation.

Reviewer #1: All comments have been addressed

Reviewer #2: All comments have been addressed

2. Is the manuscript technically sound, and do the data support the conclusions?

Reviewer #1: Yes

Reviewer #2: Yes

3. Has the statistical analysis been performed appropriately and rigorously? 

Reviewer #1: Yes

Reviewer #2: N/A

4. Have the authors made all data underlying the findings in their manuscript fully available?

Reviewer #1: Yes

Reviewer #2: Yes

5. Is the manuscript presented in an intelligible fashion and written in standard English?

Reviewer #1: Yes

Reviewer #2: Yes

6. Review Comments to the Author

Reviewer #1: Dear authors and editor,

The authors have shortenen the article as suggested.

It is present form the article is well structured, written and could be published.

Reviewer #2: I want to thank the authors for their hard work on this article. I would also like to suggest to them that "qualitative methods" refers to certain data collection modalities. There are appropriate statistics to use with qualitative data. Also, their adoption of an entirely narrative form of reporting does little to sharpen or clarify their findings. The article would have more punch if it were less discursive and more focused on the findings. However, I now feel I'm stepping into the role of advisor, which is not necessarily the job of a reviewer.

7. PLOS authors have the option to publish the peer review history of their article (what does this mean?). If published, this will include your full peer review and any attached files.

Reviewer #1: **Yes: **Cristina López de Subijana

Reviewer #2: No

---

## [Editor Report · Acceptance letter]

28 Sep 2021

PONE-D-21-11248R2 

The challenges arising from the COVID-19 pandemic and the way people deal with them. A qualitative longitudinal study. 

Dear Dr. Adamczyk:

I'm pleased to inform you that your manuscript has been deemed suitable for publication in PLOS ONE. Congratulations! Your manuscript is now with our production department. 

Kind regards, 

on behalf of

Dr. Shah Md Atiqul Haq 

Academic Editor

PLOS ONE